# Designing by Training: Acceleration Neural Network for Fast High-Dimensional Convolution

**Longquan Dai**
School of Computer Science and Engineering
Nanjing University of Science and Technology
dailongquan@njust.edu.cn

**Liang Tang**
CASA Environmental Technology Co., Ltd
CASA EM&EW IOT Research Center
tangl@casaet.com

**Yuan Xie**
Institute of Automation
Chinese Academy of Sciences
yuan.xie@ia.ac.cn

**Jinhui Tang**[*]
School of Computer Science and Engineering
Nanjing University of Science and Technology
jinhuitang@njust.edu.cn

## Abstract

The high-dimensional convolution is widely used in various disciplines but has a serious performance problem due to its high computational complexity. Over the decades, people took a handmade approach to design fast algorithms for the Gaussian convolution. Recently, requirements for various non-Gaussian convolutions have emerged and are continuously getting higher. However, the handmade acceleration approach is no longer feasible for so many different convolutions since it is a time-consuming and painstaking job. Instead, we propose an Acceleration Network (AccNet) which turns the work of designing new fast algorithms to training the AccNet. This is done by: 1, interpreting splatting, blurring, slicing operations as convolutions; 2, turning these convolutions to $g$CP layers to build AccNet. After training, the activation function $g$ together with AccNet weights automatically define the new splatting, blurring and slicing operations. Experiments demonstrate AccNet is able to design acceleration algorithms for a ton of convolutions including Gaussian/non-Gaussian convolutions and produce state-of-the-art results.

## 1 Introduction

The high-dimensional convolution undoubtedly is a common and elementary computation unit in machine learning, computer vision and computer graphics. Krähenbühl and Koltun [2011] conducted efficient message passing in the fully connected CRFs inference by the high-dimensional Gaussian convolution. Elboer et al. [2013] expressed the generalized Laplacian distance for visual matching as cascaded convolutions. Paris and Durand [2009] converted the bilateral filter [Tomasi and Manduchi, 1998] into convolution in an elevated high-dimensional space. However, the computational complexity for a $d$-D convolution (1) is proportional to $r^d$, where $r$ denotes the radius of the box filtering window $\Omega$, $\mathcal{K}_{pq}$ represents the weight between $p$ and $q$, $\mathcal{I}_p$ and $\mathcal{I}'_p$ are the values of input $\mathcal{I}$ and output $\mathcal{I}'$ at $p$. Therefore the running cost will become unacceptable for large $r$ or $d$.

$$\mathcal{I}'_p = (\mathcal{K} * \mathcal{I})_p = \sum_{q \in \Omega_p} \mathcal{K}_{pq} \mathcal{I}_q \tag{1}$$

A lot of work was devoted to solving the computational shortcoming. But most of them focus on the Gaussian filtering. This is because not only the Gaussian convolution itself serves as building blocks

---

[*]Corresponding Author.

for many algorithms [Baek and Jacobs, 2010, Yang et al., 2015] but also its acceleration approaches play important roles in defocus [Barron et al., 2015], segmentation [Gadde et al., 2016], edge-aware smoothing [Barron and Poole, 2016], video propagation [Jampani et al., 2017].

In the literature, the most popular Gaussian blur acceleration algorithm should be the Splatting-Blurring-Slicing pipeline (SBS), which is first proposed by Paris and Durand [2006], Adams et al. [2010] coined its current name. We attribute its success to *data reduction* and *separable blurring*. In SBS, pixels are "splatted"(downsampled) onto the grid to reduce the data size, then those vertexes are blurred, finally the filtered values for each pixel are produced via "slicing"(upsampling). Due to the separable blurring kernel, the $d$-D Gaussian blurring performed on those vertexes can be deemed as a sum of separable 1-D filters [Szeliski, 2011] and therefore the computational complexity per pixel is reduced from $O(r^d)$ to $O(rd)$. As the filtering window becomes small after splatting, the computational complexity can be roughly viewed as $O(d)$ which is irrelevant to the radius $r$.

According to our investigation SBS has two problems: *1, how to approximate non-Gaussian blur?* SBS is designed for the Gaussian convolution. However, the requirements for non-Gaussian blurs emerge from local Laplacian filtering [Aubry et al., 2014] and mean-field inference [Vineet et al., 2014] recently. *2, how to improve the approximation error?* Previous SBS based methods just claim that their results are good approximations for the Gaussian filtering and prove this by experiments. Since current SBS has drawbacks, how can we generalize SBS to get a better result?

We recast SBS as a neural network (AccNet) to address above two problems in this paper. The benefits are threefold: 1, our AccNet offers a unified perspective for SBS based acceleration methods; 2, the layer weights together with the activation function $g$ define the splatting, blurring and slicing convolution. So we can easily derive new splatting, blurring and slicing operations from the trained network for arbitrary convolutions. This ability entitles our network the End-to-End feature; 3, the optimal approximation error is guaranteed by AccNet in training.

## 2 Related Work

Few papers discussed acceleration algorithms for general high-dimensional convolution. Szeliski [2011] recorded a separable filtering method by SVD. Extending SVD to high-dimensional cases, we can generalize the separable filtering to high-dimensional convolution. In bilateral filtering literatures [Chaudhury and Dabhade, 2016, Dai et al., 2016], shiftable functions are exploited to approximate 1-D range kernels. This technique can also be extended to high-dimensional cases via outer product. However, its approximation terms are same as the separable filtering method and will exponentially increase with the dimension.

Current interest for fast high-dimensional convolution algorithms limits to Gaussian blur. Greengard and Strain [1991] provided the first fast Gaussian blur algorithm. Since the inception of the bilateral filter (BF) [Tomasi and Manduchi, 1998], the study for fast Gaussian convolution emerges in computer vision and computer graphics. Durand and Dorsey [2002] computed intermediate filtered images and synthesized final results by interpolation. The same approach was adopted by Porikli [2008] and Yang et al. [2009]. Paris and Durand [2006] implemented the first SBS which hints at more general approaches (bilateral grid and permutohedral lattice).

The bilateral grid [Chen et al., 2007] is a dense data structure that voxelizes the input space into a regular hypercubic lattice. By embedding inputs within the discretized space (splatting), they mix the values with a conventional Gaussian blur (blurring). The output image is extracted by resampling back into image space (slicing). The permutohedral lattice [Adams et al., 2010] is a sparse lattice that tessellates the space with simplices. By exploiting the fact that the number of vertices in a simplex grows slowly, it avoids the exponential growth of runtime that the bilateral grid suffers.

## 3 Design by Training for Fast High-Dimensional Convolution

Different from traditional output-focused neural networks, our AccNet implements the design-by-training strategy to automatically produce fast convolution pipeline and thus only interests in the activation function and weights as they define new splatting, blurring and slicing operations. In the following sections, we discuss how to transform the SBS into an AccNet as well as extensions for AccNets.

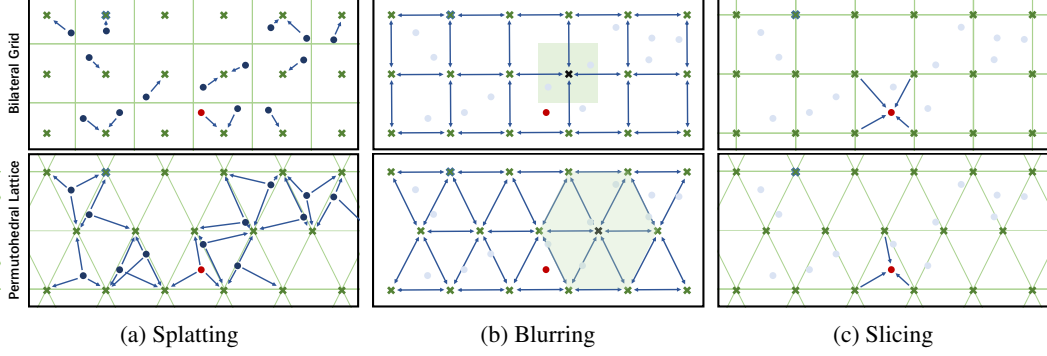

Figure 1: The splatting-blurring-slicing pipeline demonstration for bilateral grid and permutohedral lattice. The bilateral grid accumulates input values on a grid and factors the Gaussian-weighted gather into a separable Gaussian blur followed by multilinear sampling. The permutohedral lattice operates uses the permutohedral lattice. Barycentric weights within each simplex are used to resample into and out of the lattice. The separable blur is conducted along each axis.

## 3.1 Splatting, Blurring and Slicing Operations as Convolutions

Splatting voxelizes the space into a regular lattice and embeds inputs within the discretized vertices of the lattice to reduce the data size. Figure 1a illustrates the splatting operation of both bilateral grid and permutohedral lattice. The bilateral grid acceleration method trades accuracy for speed by accumulating constant values. The permutohedral lattice acceleration algorithm uses barycentric weights within each simplex to resample into the lattice. So the value of each vertice is the weighted sum of its nearby inputs. That is to say, the splatting operation conducts convolutions with a stride of $s$. Here $s$ denotes the interval of lattice vertices.

Slicing as illustrated in Figure 1c is the inverse operation of splatting. SBS employs it to synthesize filtering results from the smoothed lattice values. The bilateral grid method does this by trilinear interpolation and the permutohedral lattice algorithm takes barycentric weights to resample out of the lattice. Since the slicing values are the weighted sum of neighbor vertices, the slicing operation equals to the convolution operation. Intuitively, slicing behaves as the deconvolution layer of the fully convolutional network [Shelhamer et al., 2017] which performs upsampling by convolution.

Blurring is an alias of convolution. In the $d$-D case, the full kernel implementation for a convolution requires $r^d$ (multiply-add) operations per pixel, where $r$ is the radius of the convolution kernel. This operation can be sped up by sequentially performing 1-D convolutions along each axis (which requires a total of $dr$ operations per pixel) if the kernel is separable. Mathematically, a separable

$$\mathcal{K} = \boldsymbol{k}_1 \circ \boldsymbol{k}_2 \cdots \circ \boldsymbol{k}_d \tag{2}$$

$$\mathcal{I}' = \mathcal{K} * \mathcal{I} = \boldsymbol{k}_1 * \boldsymbol{k}_2 \cdots \boldsymbol{k}_d * \mathcal{I} \tag{3}$$

kernel $\mathcal{K}$ is the rank-one tensor (the outer product of $d$ vectors $\{\boldsymbol{k}_n, n = 1, \ldots, d\}$ (2)). Then the convolution with $\mathcal{K}$ becomes (3). For arbitrary kernels, we can reformulate it as the sum of rank-one tensors by Canonical Polyadic (CP) decomposition [Sidiropoulos et al., 2017]. In this way, we have (4) and the computational complexity per pixel for the $d$-D case becomes $O(Ndr)$. Note that the

$$\mathcal{K} = \sum_{i=1}^{N} w_i \circ \boldsymbol{k}_1^i \circ \boldsymbol{k}_2^i \cdots \circ \boldsymbol{k}_d^i \tag{4}$$

$$\mathcal{I}' = \mathcal{K} * \mathcal{I} = \sum_{i=1}^{N} w_i \cdot \boldsymbol{k}_1^i * \boldsymbol{k}_2^i \cdots * \boldsymbol{k}_d^i * \mathcal{I} \tag{5}$$

smoothing window usually is small after splatting, the computational complexity can be viewed as $O(Nd)$ which is irrelevant to $r$.

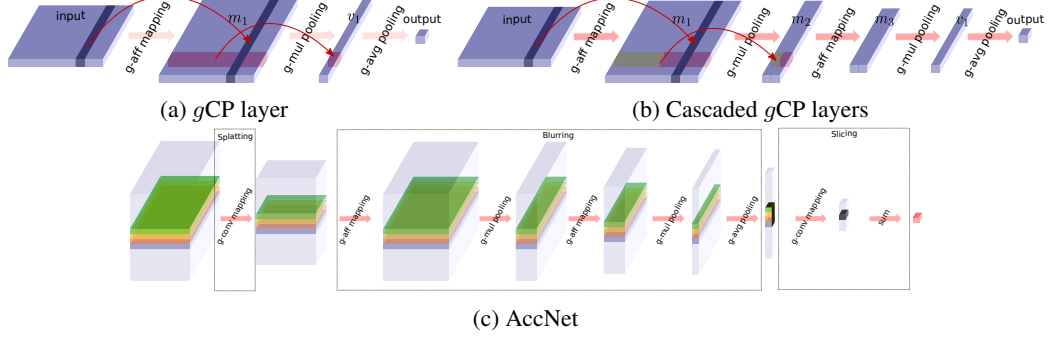

(a) gCP layer         (b) Cascaded gCP layers

(c) AccNet

Figure 2: Demonstration for $g$CP layer, cascaded $g$CP layers and AccNet. The inputs of (a) (b) are matrices formed by $[l_1^{p_i}, \ldots, l_d^{p_i}]$ (refer to section 3.2.2) and their outputs are scalars. The color cube in (c) stands for $\mathcal{L}^j$ (refer to section 3.2.4) and the color slice in the cube represents $L_{p_i}^j$, where the outputs of (a)(b)(c) are scalars, the stripes in (a)(b) and the slices in (c) with the same color present the input-output relationship.

## 3.2    Design by Training Acceleration Network (AccNet)

Essentially, our design-by-training approach is to decompose the filtering kernel (a tensor) by neural networks because a convolution can be fast computed according to (5) once (4) is obtained. In both equations, basic building blocks are multiplication and addition. If one of them is substituted by other operations, we obtain new CP decomposition (4) and separable convolution (5). That is to say, we get new kinds of splatting, blurring and slicing operations. In this section we follow the way of Cohen and Shashua [2016] to generalize (4) to $g$CP decomposition and provide corresponding $g$-convolution. The $g$CP layer and cascaded $g$CP layers are proposed for $g$CT and $g$HT decompositions.

### 3.2.1    $g$CP ($g$ Canonical Polyadic) Decomposition & $g$-Convolution

In (4) each element $\mathcal{K}_{j_1,j_2,\ldots,j_d}$ is formulated as $\sum_{i=1}^{N} w_i k_{1,j_1}^i k_{2,j_2}^i \cdots k_{d,j_d}^i$. Assuming the activation function $g : \mathbb{R} \times \mathbb{R} \to \mathbb{R}$ denotes multiplication, we have $k_{1,j_1}^i k_{2,j_2}^i \cdots k_{d,j_d}^i = k_{1,j_1}^i \times_g k_{2,j_2}^i \times_g \cdots \times_g k_{d,j_d}^i$, where $a \times_g b = g(a,b) = ab$. Let $g$ be an activation function such that $\forall a,b,c \in \mathbb{R} : g(g(a,b),c) = g(a,g(b,c)), g(a,b) = g(b,a)$, the tensor decomposition (4) can be generalized by defining $\mathcal{K}_{j_1,j_2,\ldots,j_d} = \sum_{i=1}^{N} w_i \times_g k_{1,j_1}^i \times_g \cdots \times_g k_{d,j_d}^i$. So we have $g$CP decomposition (6), where $\circ_g$ denotes the generalized outer product by replacing multiplication with the activation function $g$.

$$\mathcal{K} = \sum_{i=1}^{N} w_i \times_g k_1^i \circ_g k_2^i \cdots \circ_g k_d^i \tag{6}$$

Further, we substitute $g$ for multiplication in (1) and obtain the $g$-convolution (7).

$$\mathcal{I}_p' = (\mathcal{K} *_g \mathcal{I})_p = \sum_{q \in \Omega_p} \mathcal{K}_{pq} \times_g \mathcal{I}_q \tag{7}$$

Applying (6) to (7), we get (8) which sequentially performs $N$ 1-D $g$-convolutions.

$$\mathcal{I}' = \mathcal{K} *_g \mathcal{I} = \sum_{i=1}^{N} w_i \times_g k_1^i *_g k_2^i \cdots *_g k_d^i *_g \mathcal{I} \tag{8}$$

### 3.2.2    $g$CP Layer as $g$CP Decomposition

$\mathcal{K}_{pq}$ and $\mathcal{I}_q$ in (7) form two $d$-order tensors. Taking $\hat{\mathcal{K}}$ and $\hat{\mathcal{I}}$ to denote them and putting (6) into (8), we have (9). Letting $l_j$ be a vector and putting $\hat{\mathcal{I}}_{v_1,\ldots,v_d} = \prod_{j=1}^{d} l_{j,v_j} = l_{j,v_1} \times_g l_{j,v_2} \times_g \cdots \times_g l_{j,v_d}$ into (9), we obtain (10) which is consisted of three operations: 1, the $g$-affine mapping ($g$-aff mapping) defined by $\sum_v k_{j,v}^i \times_g l_{j,v}$; 2, the $g$-multiplication pooling ($g$-mul pooling) described by $\prod_{j=1}^{d}$; 3, the weighted average pooling ($g$-avg pooling) given by $\sum_{i=1}^{N} w_i$. The activation function $g$ introduces

$$\mathcal{I}'_{\boldsymbol{p}} = (\mathcal{K} *_g \mathcal{I})_{\boldsymbol{p}} = \sum_{j_1,\ldots,j_d} \hat{\mathcal{K}}_{j_1,\ldots,j_d} \times_g \hat{\mathcal{I}}_{j_1,\ldots,j_d}$$

$$= \sum_{v_1,\ldots,v_d} \sum_{i=1}^{N} w_i \times_g \prod_{j=1}^{d} \boldsymbol{k}^i_{j,v_j} \times_g \hat{\mathcal{I}}_{v_1,\ldots,v_d} \tag{9}$$

$$\hat{\mathcal{I}}'_{\boldsymbol{p}} = \sum_{i=1}^{N} w_i \times_g \prod_{j=1}^{d} \sum_{v} \boldsymbol{k}^i_{j,v} \times_g \boldsymbol{l}_{j,v} \tag{10}$$

nonlinearity to the three operations. Figure 2a plots the architecture, where the input is a matrix, the $g$-aff mapping transforms $\boldsymbol{l}_{j,v}$ denoted by the black line in the input to a new black vector in $\boldsymbol{m}_1$, the $g$-mul pooling maps each red vector in $\boldsymbol{m}_1$ to a scaler in vector $\boldsymbol{v}_1$ and the $g$-avg pooling reduces the element number of $\boldsymbol{v}_1$ to 1. In fact, the three operations belong to two categories. the $g$-avg pooling just is a special case of $g$-aff mapping. At last, we coin this layer as $g$CP layer as it implements the $g$CP decomposition for $\mathcal{K}$.

### 3.2.3 Cascaded $g$CP Layers as $g$HT ($g$ Hierarchical Turker) Decomposition

The expressive power of neural network has a close connection with the depth of layers. We cascade multiple $g$CP layers to extend the expressive ability in this section. The $g$CP layer maps a matrix to a scalar. As illustrated in Figure 2a, the $g$-aff mapping changes the element number of each red fiber, the $g$-mul pooling reduces the number of channels to 1 and the $g$-avg pooling decreases the element number of $\boldsymbol{v}_1$ to 1. If we replace the global pooling in the $g$-mul pooling by the local pooling, the output will become a matrix. Similarly, if we increase the output number of the last operation (the $g$-avg pooling is turned to the $g$-aff mapping), the output will be a vector. In this way, the $g$CP layer maps a matrix to another matrix and we can cascade two CP-layers together. Figure 2b provides a demo of two cascaded $g$CP layers, where the last $g$-aff mapping of the first $g$CP layer and the first $g$-aff mapping of the second $g$CP layer are merged as one $g$-aff mapping.

Cascaded $g$CP layers implement $g$ hierarchical tucker decomposition [Hackbusch and Kühn, 2009], which replaces the multiplication by $g$ in hierarchical tucker decomposition as we do for $g$CP. For example, a $g$ hierarchical turker decomposition for a 4-order tensor $\mathcal{K}$ with two layers can

$$\mathcal{K} = \sum_{m=1}^{N_2} w_m \times_g \prod_{n=1}^{2} \mathcal{K}^m_n \quad \text{with} \quad \mathcal{K}^m_n = \sum_{i=1}^{N_1} w^m_{ni} \times_g \prod_{j=1}^{2} \boldsymbol{k}^{mi}_{nj} \tag{11}$$

be expressed as (11). Put (11) into convolution formula, we have (12). Comparing (12) to the

$$\mathcal{I}'_{\boldsymbol{p}} = \sum_{m=1}^{N_2} w_m \times_g \prod_{n=1}^{2} \hat{\mathcal{I}}^m_n \quad \text{with} \quad \hat{\mathcal{I}}^m_n = \sum_{i=1}^{N_1} w^m_{ni} \times_g \prod_{i=1}^{2} \sum_{v} \boldsymbol{k}^{mi}_{nj,v} \times_g \boldsymbol{l}_{j,v} \tag{12}$$

architecture in Figure 2b, we can find that the operators $\sum_v \boldsymbol{k}^{mi}_{nj,v}$, $\prod_{i=1}^{2}$, $\sum_{i=1}^{N_1} w^m_{ni}$, $\prod_{n=1}^{2}$ and $\sum_{m=1}^{N_2} w_m$ corresponds the first $g$-aff mapping and $g$-mul pooling, the second $g$-aff mapping and $g$-mul pooling, the $g$-avg pooling, respectively.

### 3.2.4 Proposed AccNet

**Input:** in sections (3.2.2) (3.2.3) we assumed $\mathcal{I}_{\boldsymbol{p}_i} = \boldsymbol{l}^{\boldsymbol{p}_i}_1 \circ_g \cdots \circ_g \boldsymbol{l}^{\boldsymbol{p}_i}_d$ and form the matrix $[\boldsymbol{l}^{\boldsymbol{p}_i}_1, \cdots, \boldsymbol{l}^{\boldsymbol{p}_i}_d]$ as the network input for each point $\boldsymbol{p}_i$. To relax this assumption, we suppose $\mathcal{I}_{\boldsymbol{p}_i} = \sum_{j=1}^{l} \mathcal{I}^j_{\boldsymbol{p}_i}$ and $\mathcal{I}^j_{\boldsymbol{p}_i} = \boldsymbol{l}^{\boldsymbol{p}_i}_{1j} \circ_g \cdots \circ_g \boldsymbol{l}^{\boldsymbol{p}_i}_{dj}$. The blurring value of each vertice $\boldsymbol{p}_i$ on the bilateral grid or permutohedral lattice depends on the values of its neighborhoods (an image batch $\mathcal{I}_{\boldsymbol{p}_i}$). For slicing, we need $m$ vertices $\boldsymbol{p}_i$ surrounding the target point to interpolate its filtering result. So total $m$ image batches $\{\mathcal{I}_{\boldsymbol{p}_i}, 1 \leq i \leq m\}$ are required to compute the results of target points encircled by $\{\boldsymbol{p}_i, 1 \leq i \leq m\}$. To synthesize filtering values of target points encircled by $\{\boldsymbol{p}_i\}$, we compose $\mathcal{L}^j$ by concatenating $\{\boldsymbol{L}^j_{\boldsymbol{p}_i} = [\boldsymbol{l}^{\boldsymbol{p}_i}_{1j}, \cdots, \boldsymbol{l}^{\boldsymbol{p}_i}_{dj}], 1 \leq i \leq m\}$ vertically. Further $\{\mathcal{L}^j, 1 \leq j \leq l\}$ are stacked together and serves as our AccNet input. Figure 2c illustrates this, where color regions denote different parts $\{\boldsymbol{L}^j_{\boldsymbol{p}_i}\}$ of $\mathcal{L}^j$ and the light cube represents the 3-order input tensor.

**Splatting:** the splatting layer conducts the strided convolution. Theoretically, the convolution kernel $\mathcal{K}$ is arbitrary. Here, we assume $\mathcal{K} = \boldsymbol{k}_1 \circ_g \cdots \circ_g \boldsymbol{k}_d$ is a rank-one tensor in AccNet due to the

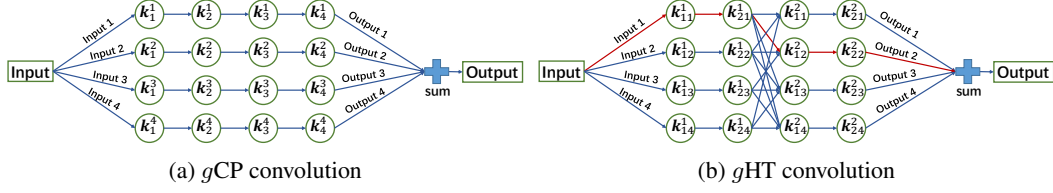

<center>(a) <em>g</em>CP convolution            (b) <em>g</em>HT convolution</center>

Figure 3: Illustration for fast filtering approaches based on $g$CP and $g$HT decompositions. Taking different tensor decomposition methods for the filtering kernel, we achieve different fast filtering algorithms. (a) plots the computation graph of fast filtering scheme (7) for the $g$CP decomposition $\mathcal{K} = \sum_{i=1}^{4} \prod_{j=1}^{4} \boldsymbol{k}_{j}^{i}$. The path indicated by arrows presents the convolution sequence with kernels $\{\boldsymbol{k}_{j}^{i}\}$. Final result is the sum of the outputs of all 4 paths. (b) shows the flow chart of fast filtering scheme (12) for the $g$HT decomposition $\mathcal{K} = \prod_{m=1}^{2} \mathcal{K}_m$ with $\mathcal{K}_m = \sum_{j=1}^{4} \prod_{i=1}^{2} \boldsymbol{k}_{ij}^{m}$. Each input connects to four outputs and thus produces four outputs. The red line indicates a convolution path. Final result is the sum of the outputs of all 16 paths.

reasons: 1, AccNet takes three layers to approximate the convolution result. Even though the splatting layer is simple, the approximation error can be reduced by increasing the complexity of the blurring layer; 2, each slice of the input tensor of the blurring layer must be a rank-one matrix and the filtering result $\mathcal{I}_{\boldsymbol{p}_i}^{\prime j}$ with a rank-one kernel $\mathcal{K}$ for a rank-one input tensor $\mathcal{I}_{\boldsymbol{p}_i}^{j} = \boldsymbol{l}_{\boldsymbol{p}_i}^{j,1} \circ_g \cdots \circ_g \boldsymbol{l}_{\boldsymbol{p}_i}^{j,d}$ is also a rank-one tensor.

**Blurring:** we prefer to employ cascaded $\boldsymbol{g}$CP-layers to compose the blurring layer of AccNet as it has more powerful expressive ability than single $\boldsymbol{g}$CP-layer. Figure 2c provides a two $\boldsymbol{g}$CP-layers example. For each slice $\boldsymbol{L}_{\boldsymbol{p}_i}^{j} = [\boldsymbol{l}_{1j}^{\boldsymbol{p}_i}, \cdots, \boldsymbol{l}_{dj}^{\boldsymbol{p}_i}]$, the blurring layer produces a scalar value $z_{\boldsymbol{p}_i}^{j}$.

**Slicing:** let $\boldsymbol{z}^j = [z_{\boldsymbol{p}_1}^{j}, \ldots, z_{\boldsymbol{p}_m}^{j}]$, the slicing layer maps $\boldsymbol{z}^j$ to a vector $\boldsymbol{t}^j$, where each element of $\boldsymbol{t}^j$ corresponds to the interpolated values of the pixels surrounded by $\{\boldsymbol{p}_i\}$ and the value of each $\boldsymbol{p}_i$ are from $\mathcal{I}_{\boldsymbol{p}_i}^{j}$. Since $\mathcal{I}_{\boldsymbol{p}_i} = \sum_{j=1}^{l} \mathcal{I}_{\boldsymbol{p}_i}^{j}$, there are total $l$ different $\boldsymbol{z}^j$ and therefore we obtain $l$ different $\boldsymbol{t}^j$. The final result is the sum of $\{\boldsymbol{t}^j, 1 \le j \le l\}$.

**$\boldsymbol{g}$ function:** the function plays an important role in our AccNet. First, it introduces nonlinearity to AccNet. This strengthens the expressive power of our AccNet. Second, it defines new convolutions. Employing $g$-conv operation, we can easily define novel splatting, blurring and slicing operations. There are many possible $\boldsymbol{g}$ functions meeting the associativity $g(g(a,b),c) = g(a,g(b,c))$ and commutativity $g(a,b) = g(b,a)$ requirements. Here we list two of them used in AccNet: 1, $g(a,b) = \max\{a,0\} \max\{b,0\}$; 2, $g(a,b) = \max\{ab,0\}$.

**Gradients:** The gradients of both sum and $g$ function can be easily obtained. Therefore, AccNet as a composition of the two basis calculations can be easily trained by the back-propagation algorithm.

## 4 Approximation & Fast Filtering

In section 3, we discussed the layers of AccNet as well as the way to transform the SBS to an AccNet. Here, we describe an approach to compose an expressive powerful AccNet and to turn it back to SBS.

**Expressive Powerful AccNet:** the expressive power of AccNet determines the approximation error. We have two ways to increase this power. One is to introduce the nonlinear activation function to AccNet. Unlike traditional SBS taking the CP decomposition for acceleration, we implement $g$CP decomposition in AccNet. Another way is to make AccNet deeper. In this way, $g$CP becomes $g$HT. At last, we note that we can choose different activation functions in different layers. This is because splatting, blurring and slicing operations are essentially convolutions therefore we can take different $g$CTs and $g$HTs to accelerate their computation.

**From AccNet to SBS:** the weights as well as the activation function of three AccNet layers define the splatting, blurring and slicing kernels. The correspondences between AccNet weights and convolution kernels are determined by (7) (8) for the $g$CP decomposition and (11) (12) for the $g$HT decomposition. For easy understanding, we visualize the computation graph of an AccNet in Figure 3. Figure 3a

Table 1: Filtering accuracy comparison for the bilateral grid acceleration method (BG), the permutohedral lattice acceleration method (PL) and our AccNet, where the sampling period of splatting is 3, the radius of blurring is 1 and the radius of original convolution is 5.

| | 2D | | | | 3D | | | | 5D | | | |
|---|---|---|---|---|---|---|---|---|---|---|---|---|
| | $\sigma = 2$ | $\sigma = 4$ | $\sigma = 8$ | $\sigma = 16$ | $\sigma = 2$ | $\sigma = 4$ | $\sigma = 8$ | $\sigma = 16$ | $\sigma = 2$ | $\sigma = 4$ | $\sigma = 8$ | $\sigma = 10$ |
| BG | 0.952 | 0.768 | 0.587 | 0.288 | 1.225 | 1.085 | 0.813 | 0.668 | 1.804 | 1.552 | 1.179 | 0.878 |
| AccNet | 0.309 | 0.249 | 0.165 | 0.054 | 0.336 | 0.276 | 0.267 | 0.171 | 0.853 | 0.465 | 0.349 | 0.259 |
| PL | 0.541 | 0.657 | 0.419 | 0.239 | 1.107 | 0.893 | 0.733 | 0.604 | 1.712 | 1.488 | 1.005 | 0.854 |
| AccNet | 0.273 | 0.175 | 0.142 | 0.051 | 0.381 | 0.243 | 0.203 | 0.153 | 0.528 | 0.423 | 0.299 | 0.213 |

takes the $g$CT decomposition $\mathcal{K} = \sum_{i=1}^{4} \prod_{j=1}^{4} \boldsymbol{k}_j^i$ to implement the fast convolution algorithm and Figure 3b records the fast convolution for the $g$HT decomposition $\mathcal{K} = \prod_{m=1}^{2} \sum_{j=1}^{4} \prod_{i=1}^{2} \boldsymbol{k}_{ij}^m$, where circles denote convolution operations with specific filtering kernels $\boldsymbol{k}$ and arrows indicate the computation order.

The two examples in Figure 3 disclose the superiority of $g$HT decomposition based acceleration algorithms. In Figure 3a, each convolution kernel is only used by one computation path. In contrast, the convolution kernel in Figure 3b is used by multiple times. The *reuse* advantages are twofold: 1, we can reduce the approximation error because more terms can be used to approximate original kernels; 2, we can reduce the execution time by reusing the convolution result sharing the same convolution node. For example, the filtering path $\boldsymbol{k}_{11}^1 \to \boldsymbol{k}_{21}^1 \to \boldsymbol{k}_{11}^2 \to \boldsymbol{k}_{21}^2$ and $\boldsymbol{k}_{11}^1 \to \boldsymbol{k}_{21}^1 \to \boldsymbol{k}_{12}^1 \to \boldsymbol{k}_{22}^2$ share the filtering results of $\boldsymbol{k}_{11}^1 \to \boldsymbol{k}_{21}^1$.

## 5 Experiments

AccNet is the first neural network producing fast convolution algorithms. To reveal its advantages, three experiments are conducted: 1, we compare our AccNet designed acceleration method to the handmade bilateral grid and permutohedral lattice acceleration methods; 2, we provide a new neural network to automatically design fast algorithm and compare it to AccNet; 3, we employ AccNet to design new acceleration algorithms for non-Gaussian convolution and demonstrate their applications. In the following experiments, the blurring layer of AccNet is composed by two cascaded $g$CP layers and the activation function is $g(a, b) = \max(ab, 0)$.

**Fast Gaussian convolution comparison:** Both bilateral grid acceleration method (BG) and permutohedral lattice acceleration method (PL) are designed for fast Gaussian convolution. The major difference between them is the underlying grid. Our AccNet can be applied to both bilateral grid and permutohedral lattice. To illustrate the filtering accuracy of the methods produced by AccNet, we keep their convolution number same to BG and PL and evaluate their filtering accuracy. Table 1 records the quantitative comparison results, where the first row denotes the dimension of the Gaussian kernel, $\sigma$ denotes the bandwidth of kernel, the accuracy is measured by MSE (the mean-square error), the first two rows record the results of BG and AccNet on the bilateral grid and the last two rows plot the results of PL and AccNet on the permutohedral lattice.

**Acceleration network comparison:** SBS sequentially conducts three convolutions. We can turn it to a CNN with three layers and further transform each CNN layer to $d$ cascaded 1-D convolution according to the CP decomposition (4) (5). The differences between this network and our AccNet are that: 1, the depth of each layer of this CNN model is proportional to the dimension of filtering kernel. In contrast, the layer depth of AccNet only depends on the desired expressive power of the layer and the expressive power of the simplest AccNet layer equal to the expressive power of CNN layer. 2, the CNN model is hard to express the $g$HT decomposition (11) as its straightforward processing pipeline is similar to Figure 3a and could not reuse intermediate results as AccNet does in Figure 3b.

The first shortcoming makes the CNN model deeper for high-dimensional convolution. We thus have to spend more time to tweak it. What's worse, the depth does not increase the expressive power of

Table 2: Two acceleration neural networks (CNN and AccNet) comparisons. The bandwidth of target Gaussian kernel is 5 and the underlying lattice is the bilateral grid.

| | 2D | | 3D | | 5D | |
|---|---|---|---|---|---|---|
| | Filtering Error | Training Time | Filtering Error | Training Time | Filtering Error | Training Time |
| CNN | 0.245 | $12.5h$ | 0.283 | $13.1h$ | 0.473 | 14h |
| AccNet | 0.239 | $7.2h$ | 0.271 | $7.3h$ | 0.461 | $7.6h$ |

this model because its expressive power is determined by the number $N$ of cascaded 1-D convolution pipelines. The second weakness causes its inferiority of the expressive power when we limits its convolution number equal to AccNet. This usually means larger filtering errors in filtering. To prove these, we plot Table 2 which records the training time as well as the filtering error measured by MSE, where the dimension of filtering kernel varies from 2-D to 5-D.

**Fast non-Gaussian filtering:** Non-Gaussian blur becomes popular recently. To illustrate the power of our AccNet, we demonstrate three applications of fast non-Gaussian filtering in machine learning, computer vision and computer graphics, respectively.

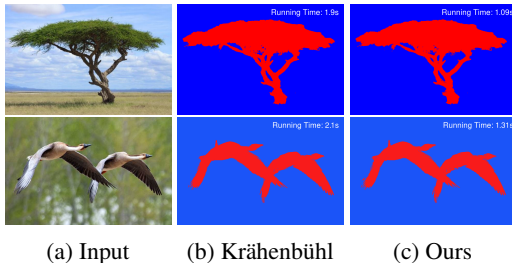

|      (a) Input      |   (b) Krähenbühl    |      (c) Ours       |

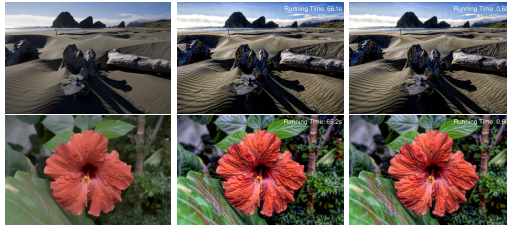

|      (a) Input      |      (b) Paris      |      (c) Ours       |

Figure 4: Pixel-level segmentation results of two fully connected CRF implementations. (a) is input images. (b) is the segmentation results of Krähenbühl. (c) records our segmentation results.

Figure 5: Detail enhancement of two local Laplace filtering implementations. (a) is input images. (b) is the filtering results of Paris. (c) denotes our results.

Table 3: Stereo matching quantitative comparison.

|  | All | | | NoOcc | | |
|---|---|---|---|---|---|---|
|  | bad 1% | MAE | RMS | bad 1% | MAE | RMS |
| [Zbontar and LeCun, 2015] | 20.07 | 5.93 | 18.36 | 10.42 | 1.94 | 9.07 |
| [Barron and Poole, 2016] | 19.49 | 2.81 | 8.44 | 11.33 | 1.40 | 5.23 |
| Ours | 19.21 | 2.13 | 7.79 | 10.41 | 1.34 | 4.96 |

*CRF inference:* The pairwise edge potentials used in the fully connected CRFs [Krähenbühl and Koltun, 2011] is the Gaussian mixture kernels. Krähenbühl and Koltun [2011] provided a highly efficient approximate inference algorithm by showing a mean field update of all variables in a fully connected CRF can be performed using Gaussian filtering in the feature space. In order to speed up the computation via the separability of the Gaussian kernel $G_i$, Krähenbühl has to perform multiple times Gaussian filtering. Employing AccNet, we can accelerate the Gaussian mixture kernels directly. Compared to the original method, we save 60% of the time while producing the same segmentation results as shown in Figure 4.

*Bilateral solver:* Bilateral solver [Barron and Poole, 2016] allows for some optimization problems with bilateral affinity terms to be solved quickly, and also guarantees that the solutions are smoothed within objects, but not smooth across edges. Although the prior used by bilateral solver is arbitrary in theory, bilateral solver can only take the Gaussian function as it is the only function can be presented by SBS before our work. Here we take the smooth exponential family prior [Zhang and Allebach, 2008] to construct non-Guassian bilateral solver and apply it stereo post-processing procedure of MC-CNN [Zbontar and LeCun, 2015] following the way of [Barron and Poole, 2016]. In Table 3, we record the quantitative results, where "bad 1%" presents the percent of pixels whose disparities are wrong by more than 1, "MAE" stands for the mean absolute error and "RMS" is the root mean square error.

*Local Laplace filtering:* Local Laplacian filter [Paris et al., 2011] is an edge-aware operator that defines the output image $\bar{I}$ by constructing its Laplacian pyramid $\{L[\bar{I}]\}$ coefficient by coefficient. Aubry et al. [2014] present the Laplacian coefficient at level $l$ and position $p$ as the nonlinear convolution $\{L_l[\bar{I}](p)\} = \sum_{q \in \Omega_p} D_l(q - p) f(I_q - g)(I_q - g)$, where $f$ is a continuous function, $D_l$ is the difference-of-Gaussians filter defining the pyramid coefficients at level $l$ and $g$ is the coefficient of the Gaussian pyramid at $(l, p)$. Obviously, this convolution can be accelerated by

AccNet and achieves speed-ups on the order of 100 times. Figure 5 visualizes the similar detail enhancement results of Paris and ours.

# 6 Conclusion

In this paper, we propose the first neural network producing fast high-dimensional convolution algorithms. We take AccNet to express the approximation function of SBS and generalize SBS by changing the architecture of AccNet. Once training is finished, new fast convolution algorithm can be easily derived from the weights and activation functions of each layer. Experiments prove the effectiveness of our algorithm.

# 7 Acknowledgment

This work was supported by the 973 Program (Project No. 2014CB347600), the National Natural Science Foundation of China (Grant No. 61701235, 61732007, 61522203, 61772275, 61873293 and 61772524), the Fundamental Research Funds for the Central Universities (Grant No. 30917011323) and the Beijing Municipal Natural Science Foundation (Grant No. 4182067).

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
