[Reviews · NeurIPS 2018]

Reviewer 1



Post Authors' Response Thanks a lot the authors for their response. I am happy with the responses they have provided to my initial concerns which have improved the manuscript. I would encourage authors to add an appendix should they believe they can convey a more complete message without having to wait for drafting another another longer manuscript. ------------------------------------------------------------ Initial Review The author/authors propose the first neural network producing fast high-dimensional convolution algorithms. They utilise the new Acceleration Network (AccNet) to express the approximation function of splatting/blurring/slicing (SBS) and generalise SBS by changing the architecture of AccNet. Upon finishing the training process, the proposed fast convolution algorithm can be derived from the weights and activation functions of each layer. They conducted various experiments that prove the effectiveness of the proposed algorithm. The papers is very well written and of good quality. The main part of the paper, i.e. 3rd to 6th page, would benefit from making clearer the exact contribution of the authors. Reason is that it sometimes provides too much information about the fundamentals and the exact implementation and adjustments are likely to be missed. Section 3.2.4 that describes the proposed algorithm could serve for this purpose, i.e. what is being added or proposed to further enhance the SBS. Figure 3 shows the fast filtering approaches based on gCP and gHT decompositions. Is this based on section 3.2.3 that has been proposed by Hackbusch and Kuehn or includes other tensor decomposition methods? I reckon the latter, but it needs to have a better connection to what is being proposed further on especially when describing the reuse of convolution kernels by multiple times. -Having said all the above, I think that the results are very interesting and demonstrate a solid computational improvement. Is Figure 5 accurate? Have you really achieved such a reduced time in the detail enhancement, i.e. 66.1s vs 0.65s. That would be a significant improvement. By the way the font size is quite small to be honest. -The weak point in my humble opinion is the few examples that are presented to demonstrate the computational efficiency, such as figures 4 and 5. I think further evaluation in larger data set is needed so that the bias can be reduced and ascertain the generalisation of this approach, as one could now claim that the examples presented have been selected to suit the proposed algorithm. -Please expand slightly the conclusion if possible, as it reads as if you have been running out of space. Highlighting the main findings and contributions will enhance the paper. Minor issues: a) Line 162, page 5 correct "approximates" to "approximate" b) Figure 3 caption second to last sentence "outputs" c) Line 273, page 8, please remove the redundant "the"

Reviewer 2



The paper proposes an idea to accelerate high-dimensional convolutions with non-gaussian kernels by approximating the splatting-blurring-slicing operations developed for gaussian convolution by learned neural networks. First of all let me point out i'm not an expert in this domain. I found some of the mathematics hard to follow, sometimes also seemingly due to poor writing. For example "g-aff mapping changes the element number of each red fiber" -- fibers are nowhere else mentioned in the paper. What fiber ? Also acronyms are profusely scattered through the paper, often way before they are defined, e.g. "gCP" is in the abstract and in the first 4 pages before in the fifth there's an attempt to define it. Experiments-wise, i think there is one problem with the paper: it's main goal - accelerating high-dimensional convolutions - does not seem to be properly evaluated; there is not a single table with a timing comparison, just an evaluation of how accurate it is. How much faster are the resulting convolutions compared to non-accelerated standard versions ? Regarding writing, it is sometimes very coloquial, for example: "a ton of convolutions ", or "Philipp has to perform multiple times Gaussian filtering", where Philipp is the an author's first name. Other remarks: - table 1: "filtering accuracy: " is not a good description of mean squared error. A reader will assume higher is better.

Reviewer 3



# Summary The work proposes a novel differentiable and fast network architecture to learn high-dimensional filtering operations. # Paper Strengths - Important problem: high-dimensional convolutions have applications for inference in CRF, computer graphics and CNN operators, among others. Computational efficiency is a big problem in this domain. - The paper provides a solid contribution towards generically accelerating these filtering operations: it applies the generalised canonical decomposition from [Cohen and Shashua, 2016] to convolutions and suggests an architecture to mimic the behaviour of established high dimensional filtering techniques. - The paper is well written; however pretty condensed. - The presented experiments cover a large range of possible questions: - comparison to related state-of-the-art: filtering defined on the bilateral grid or the permutohedral lattice - comparison to baselines: a CNN that implements splat/blur/slice with canonical polyadic decompositions - applications to inference in dense CRFs: semantic segmentation and stereo - a computer graphics application # Paper Weaknesses - Some parts of the work are harder to follow and it helps to have checked [Cohen and Shashua, 2016] for background information. # Typos and Presentation - The citation of Kraehenbuehl and Koltun: it seems that the first and last name of the first author, i.e. Philipp, are swapped. - The paper seems to be using a different citation style than the rest of the NIPS submission. Is this intended? - line 111: it might make sense to not call g activation function, but rather a binary operator; similar to Cohen and Shashua, 2016. They do introduce the activation-pooling operator though that fulfils the required conditions. - line 111: I believe that the weight w_i is missing in the sum. - line 114: Why not mention that the operator has to be associative and commutative? - eq 6 and related equations: I believe that the operator after w_i should be the multiplication of the underlying vector space and not \cross_g: It is an operator between a scalar and a tensor, and not just between two scalars. - line 126: by the black *line* in the input # Further Questions - Would it make sense to include and learn AccNet as part of a larger predictor, e.g., for semantic segmentation, that make use of similar operators? - Do you plan to publish their implementation of the proposed AccNet? # Conclusion The work shows that the proposed method is expressive enough to approximate high-dimensional filtering operations while being fast. I think the paper makes an interesting contribution and I would like to see this work being published.